# Synthesis, Structure, and Characterizations of a Heterobimetallic Heptanuclear Complex [Pb$_2$Co$_5$(acac)$_{14}$]

Yuxuan Zhang, Zheng Wei and Evgeny V. Dikarev *

Department of Chemistry, University at Albany, State University of New York, 1400 Washington Ave, Albany, NY 12222, USA; yzhang50@albany.edu (Y.Z.); zwei@albany.edu (Z.W.)
* Correspondence: edikarev@albany.edu

**Abstract:** An unusual heterobimetallic volatile compound [Pb$_2$Co$_5$(acac)$_{14}$] was synthesized by the gas phase/solid-state technique. The preparation can be readily scaled up using the solution approach. X-ray powder diffraction, ICP-OES analysis, and DART mass spectrometry were engaged to confirm the composition and purity of heterobimetallic complex. The composition is unique among the large family of lead(tin): transition metal = 2:1, 1:1, and 1:2 $\beta$-diketonates compounds that are mostly represented by coordination polymers. The molecular structure of the complex was elucidated by synchrotron single crystal X-ray diffraction to reveal the unique heptanuclear moiety {Co(acac)$_2$[Pb(acac)$_2$-Co(acac)$_2$-Co(acac)$_2$]$_2$} built upon bridging interactions of acetylacetonate oxygens to neighboring metal centers that bring their coordination numbers to six. The appearance of unique heptanuclear assembly can be attributed to the fact that the [Co(acac)$_2$] units feature both *cis*- and *trans-bis*-bridging modes, making the polynuclear moiety rather flexible. This type of octahedral coordination is relatively unique among known lead(tin)-3$d$ transition metal $\beta$-diketonates. Due to the high-volatility, [Pb$_2$Co$_5$(acac)$_{14}$] can be potentially applied as a MOCVD precursor for the low-temperature preparation of lead-containing functional materials.

**Keywords:** single crystal; lead complexes; cobalt complexes; heterometallic structures; heptanuclear molecule

## 1. Introduction

A number of divalent lead-transition metal heterobimetallic compounds Pb$_x$M$_y$($\beta$-dik)$_{2x+2y}$ (M = Mn, Co, Ni, Fe, Zn) have been synthesized as single-source precursors (SSPs) for functional lead-containing transition-metal oxides or fluorides [1–3]. $\beta$-Diketonate ligands such as acetylacetonate (*acac*) and fluorinated hexafluoroacetylacetonate (*hfac*) and trifluoroacetylacetonate (*tfac*) have been employed to form heterometallic complexes that exhibit good volatility and solubility. More importantly, the ligands endowed the above SSPs with low-temperature decomposition to produce target materials such as magnetoelectrics Pb$_2$MF$_6$ (M = Ni and Co) [1,4] or multiferroics Pb$_2$Fe$_2$O$_5$ [3,5]. Most of the lead-transition metal $\beta$-diketonates appeared to have polymeric structures built upon combination of Pb($\beta$-dik)$_y$ ($y$ = 0, 1, 2) and M($\beta$-dik)$_x$ ($x$ = 2, 3) units connected by Lewis acid–base Pb–O/M–O interactions (bridging bonds) with three different Pb:M ratios of 2:1, 1:1, and 1:2 [1–3]. Within the heterometallic assemblies, lead and transition metals both exhibit coordination numbers of six with the former preferring *cis*-bridging and the latter *trans*-bridging with neighboring fragments [1–3,6–14]. In this work, we describe the synthesis and characterization of a new lead-containing heterobimetallic heptanuclear homoleptic complex [Pb$_2$Co$_5$(acac)$_{14}$] which is distinguished from the previously reported Pb–M compounds in several aspects. The ratio of Pb:Co = 2:5 is unique, while the heptanuclear structure is constructed by both *cis*- and *trans*-bridged [Co(acac)$_2$] units providing more complex/flexible coordination environment. Due to the good volatility, the [Pb$_2$Co$_5$(acac)$_{14}$] complex is interesting as a prospective precursor in the metal–organic chemical vapor deposition (MOCVD) for

the preparation of high-technological thin films with desired stoichiometry [15–17]. The unusual lead:cobalt ratio makes it attractive to investigate the potential application for the low-temperature formation of lead–cobalt-coated anodes, lone-pair multiferrorics, and superconducting materials [18–27].

## 2. Materials and Methods

### 2.1. Materials and Measurements

Lead(II) acetylacetonate [Pb(acac)$_2$] and cobalt(II) acetylacetonate [Co(acac)$_2$] were purchased from Sigma-Aldrich and used as received after checking their powder X-ray diffraction patterns. The ICP-OES analysis was carried out on ICPE-9820 plasma atomic emission spectrometer, Shimadzu. The IR spectrum was measured using IRTracer-100 Fourier Transform Infrared Spectrophotometer, Shimadzu. The DART-MS spectra were recorded on AccuTof 4G LC-plus DART mass spectrometer, JEOL. X-ray powder diffraction data were collected on a Rigaku MiniFlex 6G benchtop diffractometer (Cu K$\alpha$ radiation, D/teX Ultra silicon strip one-dimensional detector, step of 0.01° 2$\theta$, 20 °C). Le Bail fit refinement for powder diffraction patterns has been performed using TOPAS, version 4 software package (Bruker AXS, Billerica, MA, USA, 2006).

### 2.2. General Synthetic Procedures

The solid-state/gas phase synthesis of heterometallic complexes was carried out by grinding and sealing the stoichiometric mixture of starting reagents in an evacuated glass ampule and placing it into a gradient furnace. Solution synthesis of heterometallic complex was performed by adding dry and deoxygenated organic solvent to the stoichiometric mixture of starting reagents in a flask under a dry, oxygen-free argon atmosphere using standard Schlenk and glove box techniques. Detailed synthetic procedures are described in Section 3.1.

### 2.3. X-ray Crystallographic Procedures

The crystals of [Pb$_2$Co$_5$(acac)$_{14}$] were immersed in cryo-oil, mounted on a glass fiber, and measured at the temperature of 100(2) K. The X-ray diffraction data were collected on a Huber Kappa system with a DECTRIS PILATUS3 X 2M(CdTe) pixel array detector using $\phi$ scans (synchrotron radiation at $\lambda$ = 0.49594 Å) located at the Advanced Photon Source, Argonne National Laboratory (NSF's ChemMatCARS, Sector 15, Beamline 15-ID-D). The crystals of [PbCo(acac)$_4$] were immersed in cryo-oil, mounted on a loop, and measured at the temperature of 100(2) K. The X-ray diffraction data were collected on a Bruker D8 SMART diffractometer using Mo K$\alpha$ radiation. The dataset reduction and integration for both structures were performed with the Bruker software package SAINT (version 8.38A) [28]. The data were corrected for absorption effects using the empirical methods as implemented in SADABS (version 2016/2) [29]. The structures were solved by SHELXT (version 2018/2) [30] and refined by full-matrix least-squares procedures using the Bruker SHELXTL (version 2019/2) [31] software package through the OLEX2 graphical interface [32]. All non-hydrogen atoms were refined anisotropically. Hydrogen atoms were included in idealized positions for the structure factor calculations with $U_{iso}$(H) = 1.2 $U_{eq}$(C) and $U_{iso}$(H) = 1.5 $U_{eq}$(C) for methyl groups. Crystallographic data and details of the data collection and structure refinement are listed in Table 1.

**Table 1.** Crystal data and structure refinement parameters for [Pb$_2$Co$_5$(acac)$_{14}$] and [PbCo(acac)$_4$].

| Compound | Pb$_2$Co$_5$(acac)$_{14}$ | PbCo(acac)$_4$ |
|---|---|---|
| CCDC | 2268578 | 2268579 |
| Empirical formula | C$_{70}$H$_{98}$Co$_5$Pb$_2$O$_{28}$ | C$_{20}$H$_{28}$CoPbO$_8$ |
| Formula weight | 2096.51 | 662.54 |
| Temperature (K) | 100(2) | 100(2) |
| Wavelength (Å) | 0.49594 | 0.71073 |
| Crystal system | Monoclinic | Monoclinic |
| Space group | $P2_1/c$ | $P2_1/c$ |
| $a$ (Å) | 23.8452(6) | 8.7792(15) |
| $b$ (Å) | 10.7063(3) | 20.142(3) |
| $c$ (Å) | 16.3561(4) | 13.720(2) |
| $\beta$ (°) | 97.0960(10) | 91.957(2) |
| $V$ (Å$^3$) | 4143.63(19) | 2424.8(7) |
| $Z$ | 2 | 4 |
| $\rho_{calcd}$ (g·cm$^{-3}$) | 1.680 | 1.815 |
| $\mu$ (mm$^{-1}$) | 1.989 | 7.657 |
| $F(000)$ | 2082 | 1284 |
| Crystal size (mm$^3$) | $0.09 \times 0.07 \times 0.04$ | $0.16 \times 0.13 \times 0.10$ |
| $\theta$ range for data collection (°) | 1.201–19.317 | 2.321–28.277 |
| Reflections collected | 129,052 | 20,649 |
| Independent reflections | 10,257 [$R_{int} = 0.0596$] | 5640 [$R_{int} = 0.0783$] |
| Transmission factors (min/max) | 0.7421/0.8414 | 0.6241/0.7563 |
| Completeness to full $\theta$ (%) | 99.6 | 99.7 |
| Data/restraints/params. | 10,257/0/490 | 5640/0/279 |
| $R1$,[a] $wR2$[b] ($I > 2\sigma(I)$) | 0.0198, 0.0533 | 0.0545/0.1296 |
| $R1$,[a] $wR2$[b] (all data) | 0.0225, 0.0542 | 0.0975/0.1496 |
| Quality-of-fit[c] | 1.064 | 1.029 |

[a] $R1 = \Sigma ||F_o| - |F_c||/\Sigma|F_o|$. [b] $wR2 = [\Sigma[w(F_o{}^2 - F_c{}^2)^2]/\Sigma[w(F_o{}^2)^2]]$. [c] Quality-of-fit = $[\Sigma[w(F_o{}^2 - F_c{}^2)^2]/(N_{obs} - N_{params})]^{\frac{1}{2}}$, based on all data.

## 3. Results and Discussion

### 3.1. Synthesis and Properties of [Pb$_2$Co$_5$(acac)$_{14}$] and [PbCo(acac)$_4$]

Synthesis of heterometallic complex [Pb$_2$Co$_5$(acac)$_{14}$] was carried out by both solid-state and solution approaches through the stoichiometric reaction (Equation (1)). For the solid-state synthesis, 10 mg of [Pb(acac)$_2$] (0.025 mmol) and 16 mg of [Co(acac)$_2$] (0.062 mmol) were sealed in an evacuated glass ampule. The ampule was placed in a furnace with a temperature gradient of 130 to 120 °C. After one day of heating, block-shaped violet [Pb$_2$Co$_5$(acac)$_{14}$] crystals were sublimed to the cold zone of the container (16 mg, ca. 62% yield). The product can be obtained in nearly quantitative yield by extending the reaction time to one week under the same conditions. Microcrystalline powder of [Pb$_2$Co$_5$(acac)$_{14}$] was synthesized by dissolving 100 mg of [Pb(acac)$_2$] (0.25 mmol) and 160 mg of [Co(acac)$_2$] (0.62 mmol) in 20 mL of dry and deoxygenated hexanes under argon atmosphere. The solution was stirred at room temperature for 1 day, resulting in a large amount of violet precipitate being formed. The solid was filtered off and dried under vacuum at 100 °C sand bath overnight to afford the final product with a practically quantitative yield.

$$2Pb(acac)_2 + 5Co(acac)_2 \rightarrow Pb_2Co_5(acac)_{14} \tag{1}$$

The block-shaped violet crystals acquired from the solid-state reaction were first checked by the ICP-OES analysis, which revealed the metal ratio of Pb:Co as 2:5, different from any previously investigated Pb$_x$M$_y$($\beta$-dik)$_{2x+2y}$ complexes [1–3]. X-ray powder diffraction was applied to check the phase purity of the [Pb$_2$Co$_5$(acac)$_{14}$] bulk products obtained from the solid state and solution methods (Figure 1). The Le Bail fit was performed to confirm that the experimental powder patterns of the bulk products correspond to the theoretical spectrum calculated from the single crystal X-ray data (Figure 1 and Table 2).

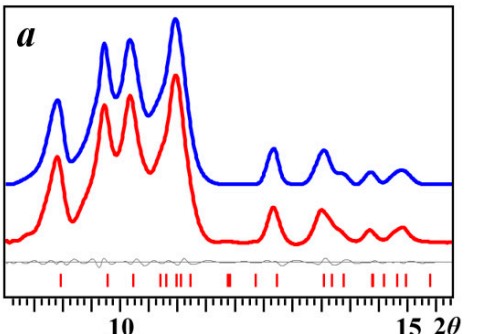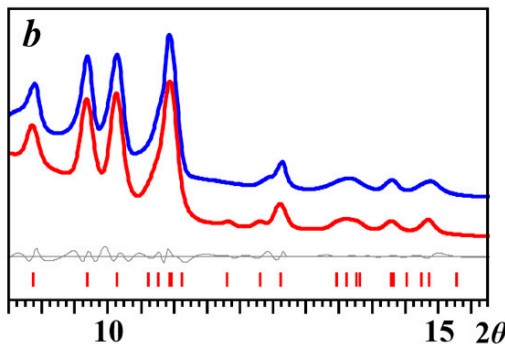

**Figure 1.** X-ray powder diffraction patterns of (**a**) [Pb$_2$Co$_5$(acac)$_{14}$] bulk product obtained from the solution reaction and the Le Bail fit; (**b**) [Pb$_2$Co$_5$(acac)$_{14}$] crystals obtained from the solid-state reaction and the Le Bail fit. In both pictures, blue and red curves represent experimental and calculated patterns, respectively. Gray is the differential curve with theoretical peak positions shown as red bars at the bottom.

**Table 2.** Comparison of the unit cell parameters for [Pb$_2$Co$_5$(acac)$_{14}$] product obtained from the single crystal refinement and the Le Bail fit.

|  | Single Crystal Data (−173 °C) | Solution Product Le Bail Fit (20 °C) | Solid-State Product Le Bail Fit (20 °C) |
|---|---|---|---|
| Space group | $P2_1/c$ | $P2_1/c$ | $P2_1/c$ |
| $a$ (Å) | 23.8452(6) | 23.939(3) | 23.933(8) |
| $b$ (Å) | 10.7063(3) | 10.9156(10) | 10.908(4) |
| $c$ (Å) | 16.3561(4) | 16.399(2) | 16.496(5) |
| $\beta$ (°) | 97.0960(10) | 96.767(11) | 97.026(13) |
| $V$ (Å$^3$) | 4143.63(19) | 4255.5(9) | 4274(2) |

Direct Analysis in Real Time (DART) mass spectrometry was performed to check the retention of heterometallic structure and analyze the ions in the gas phase. In the positive mode mass spectrum of [Pb$_2$Co$_5$(acac)$_{14}$] (SI, Figure S1 and Table S1) heterometallic ion peaks such as [PbCo$_3$(acac)$_7$]$^+$, [PbCo$_4$(acac)$_9$]$^+$, and [Pb$_2$Co(acac)$_5$]$^+$ that represent the fragmentation of the heptanuclear [Pb$_2$Co$_5$(acac)$_{14}$] molecule can be clearly detected. All heterometallic ion fragments with relatively high intensities are measured precisely with the difference between measured and calculated *m/z* being smaller than 0.006 and characteristic isotope distribution patterns (SI, Figure S1 and Table S1). The peak corresponding to heptanuclear ion [M-L]$^+$ (M = [Pb$_2$Co$_5$(acac)$_{14}$], L = *acac*) was not detected in the DART mass spectrum, indicating that the structure is quite fragile. In fact, this is the first heterometallic molecule in our research that was found not to feature the [M-L]$^+$ peak in the positive mode, though we should notice that it is the heaviest so far to be investigated. At the same time, the fragment peaks unambiguously confirm the (+2) oxidation states of both lead and cobalt.

The IR spectrum of [Pb$_2$Co$_5$(acac)$_{14}$] was also recorded and shown in the SI, Figure S2. The [Pb$_2$Co$_5$(acac)$_{14}$] complex is stable in the presence of oxygen in the solid state, but is quite sensitive to moisture. It has a good solubility in polar, weakly/non-coordinating solvents such as CHCl$_3$ and CH$_2$Cl$_2$ at room temperature, but shows a poor solubility in non-polar solvents such as hexanes, pentanes, cyclohexane, and toluene at room temperature. [Pb$_2$Co$_5$(acac)$_{14}$] can be quantitatively sublimed at as low as 105 °C under static vacuum conditions (sealed evacuated ampule) and starts to show the traces of decomposition when the temperature is raised to 150 °C.

The synthesis of heterometallic complex [PbCo(acac)$_4$] was carried out by the solid-state approach. First, 15 mg of [Pb(acac)$_2$] (0.037 mmol) and 9 mg of [Co(acac)$_2$] (0.035 mmol) were sealed in an evacuated glass ampule. The ampule was placed in a furnace with a temperature gradient of 115 to 105 °C. After one day of heating, block-shaped pink

crystals of [PbCo(acac)$_4$] were sublimed to the "cold" zone of the container (10 mg, ca. 42% yield). The differences in synthetic procedures for the preparation of [Pb$_2$Co$_5$(acac)$_{14}$] and [PbCo(acac)$_4$] are the ratio of starting reagents and the temperature in the "hot" zone of the ampule. The crystal colors of two compounds are noticeably different. [PbCo(acac)$_4$] is structurally analogous to the previously reported [PbM($\beta$-dik)$_4$] compounds (M = Mn, Fe, and Zn; $\beta$-dik = *acac*, *tfac*, and *hfac*) [1–3] and was obtained for comparison of bond distances and coordination mode of Co ion, which are discussed in detail in Section 3.2.

### 3.2. Single Crystal Structure of [Pb$_2$Co$_5$(acac)$_{14}$]

In this work, we used sterically uncongested $\beta$-diketonate ligand, acetylacetonate (*acac*), to synthesize a hetero*bi*metallic assembly [Pb$_2$Co$_5$(acac)$_{14}$], which contains both *cis*- and *trans*-bridging [Co(acac)$_2$] units. Single-crystal X-ray diffraction analysis revealed that the solid-state structure of this heterometallic complex contains discrete heptanuclear molecules with metal chain in an order of [Co-Co-Pb-Co-Pb-Co-Co]. The central Co1 atom (Figure 2) sits on an inversion center; therefore, there are only four crystallographically independent metal sites (three Co and one Pb) within the structural motif. Each metal center has two chelating *acac* ligands and fulfills its coordination environment by two additional contacts with oxygen atoms that are chelating its neighbors. The [M(acac)$_2$] (M = Co, Pb) groups are connected through Lewis acid–base M–O interactions of 2.09–2.26 and 2.81–2.86 Å for Co and Pb ions, respectively (Tables 3 and 4), that are significantly shorter than the sum of the corresponding van der Waals radii. Although all metal centers have a coordination number of 6, the coordination environments are distinctively different. Two outmost [Co(acac)$_2$] fragments offer only one *acac* oxygen each for the bridging interactions with the neighboring [Co(acac)$_2$] units, which, in turn, offer two *acac* oxygens for bridging interactions with the end-chain [Co(acac)$_2$] units in a *cis*-fashion and an additional *acac* oxygen to bridge the [Pb(acac)$_2$] unit. The bond distances for bridging Co–O bonds in these units are between 2.09 and 2.22 Å, longer than chelating and chelating–bridging Co–O distances in the structural motif (Table 3). The cobalt atoms in these two units maintain a slightly distorted octahedral geometry with two *acac* ligand planes being almost perpendicular to each other. Since both cobalt centers in the [Co(acac)$_2$] units appear as *bis-cis*-chelated, they are obviously chiral. The Co2 and Co3 exhibit the same chirality (either $\Delta,\Delta$ or $\Lambda,\Lambda$). Considering the inversion center in the middle of this heptanuclear assembly, the compound is *meso* and does not feature diastereomers. In the central [Co(acac)$_2$] fragment, Co1 ion has two chelating *acac* ligands that are located in a plane, with two bridging Co–O interactions from the neighboring [Pb(acac)$_2$] units in a *trans*-configuration. Both *acac* oxygens from one of the ligand in the central [Co(acac)$_2$] group are pure chelating, while those from the second ligand bridge to [Pb(acac)$_2$] groups on both sides. Since these two *trans*-bridging Co–O distances are much longer than those four from chelating bonds, the coordination of the central Co ion can be described as an axially elongated octahedral geometry. In the heptanuclear structure, the [Pb(acac)$_2$] units act as a glue to connect two [Co$_2$(acac)$_4$] fragments on both sides with the central [Co(acac)$_2$] unit by two bridging interactions through one oxygen from each of two *acac* ligands. The $6s^2$ lone electron pair of the lead repels two Pb-chelating *acac* ligands to be in a face-to-face fashion arrangement with a dihedral angle of 33.01°.

The appearance of both *cis*- and *trans*-bridging modes for the [Co(acac)$_2$] units in [Pb$_2$Co$_5$(acac)$_{14}$] makes it unique among polynuclear cobalt and heterometallic lead–cobalt $\beta$-diketonate complexes. Thus, the tetranuclear structure of [Co$_4$(acac)$_8$] (Figure 3) features only the *cis*-bridging mode to combine four [Co(acac)$_2$] monomers together [6]. In contrast, the polymeric heterometallic structure of [PbCo(acac)$_4$] (Figure 4) shows only *trans*-bridged [Co(acac)$_2$] building up the 1D polymeric chain with [Pb(acac)$_2$] units in 1:1 ratio. Analogous $\beta$-diketonate [PbCo(hfac)$_4$] [1] also has a polymeric structure but it is constructed by alternating [Co(hfac)$_3$]$^-$ and [Pb(acac)]$^+$ units. In turn, all [Pb(acac)$_2$] fragments in the lead-transition metal $\beta$-diketonate structures [1–3] are *cis*-bridging.

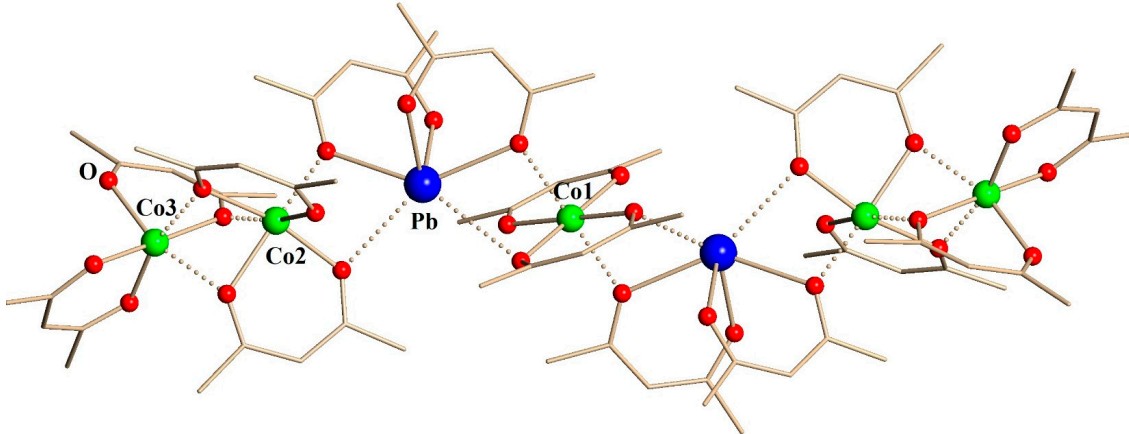

**Figure 2.** Molecular structure of [Pb$_2$Co$_5$(acac)$_{14}$] complex. Hydrogen atoms are omitted for clarity. All bridging M–O bonds are shown as dotted lines. Full view of molecular structure can be found in the SI, Figure S3.

**Table 3.** Co–O bond distances (Å) in the structures of [Pb$_2$Co$_5$(acac)$_{14}$] and in other cobalt-containing polynuclear/heterometallic complexes.

| | **Co–O$_c$** | **Co–O$_{c\text{-}b}$** | **Co–O$_b$ (*cis*)** | **Co–O$_b$ (*trans*)** |
|---|---|---|---|---|
| Pb$_2$Co$_5$(acac)$_{14}$ | 1.98–2.05 | 2.02–2.08 | 2.09–2.22 | 2.26 |
| Co$_4$(acac)$_8$ | 2.00–2.03 | 2.02–2.11 | 2.09–2.20 | |
| PbCo(acac)$_4$ | 2.02–2.03 | 2.03–2.04 | | 2.22–2.24 |
| PbCo(hfac)$_4$ | 2.05 | 2.04–2.07 | | |

c—chelating; c-b—chelating-bridging; b—bridging.

**Table 4.** Pb–O bond distances (Å) in the structures of [Pb$_2$Co$_5$(acac)$_{14}$] and other lead–cobalt *β*-diketonate complexes.

| | **Pb–O$_c$** | **Pb–O$_{c\text{-}b}$** | **Pb–O$_b$ (*cis*)** |
|---|---|---|---|
| Pb$_2$Co$_5$(acac)$_{14}$ | 2.34–2.35 | 2.48–2.49 | 2.81–2.86 |
| PbCo(acac)$_4$ | 2.31–2.32 | 2.51–2.52 | 2.86–2.89 |
| PbCo(hfac)$_4$ | 2.32 | | 2.74–2.81 |

c—chelating; c-b—chelating-bridging; b—bridging.

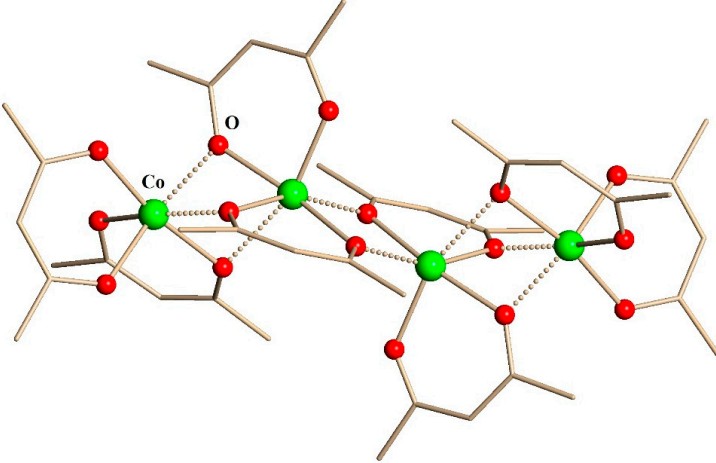

**Figure 3.** Molecular structure of tetranuclear [Co$_4$(acac)$_8$]. Hydrogen atoms are omitted for clarity. All bridging Co–O bonds are shown as dotted lines.

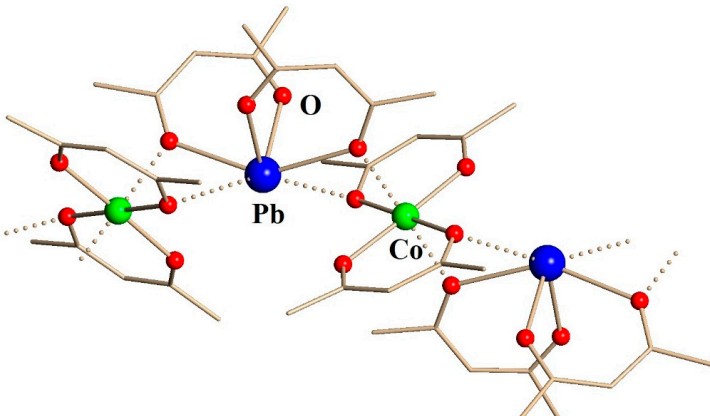

**Figure 4.** Fragment of crystal structure of polymeric [PbCo(acac)$_4$] compound. Hydrogen atoms are omitted for clarity. All bridging M–O bonds are shown as dotted lines. Full view of the structure can be found in the SI, Figure S4.

## 4. Conclusions

The hetero*bi*metallic complex [Pb$_2$Co$_5$(acac)$_{14}$] was synthesized by both solid-state and solution techniques and its unique heptanuclear molecular structure was revealed by synchrotron single-crystal X-ray diffraction. It was shown that the [Co(acac)$_2$] units in the assembly exhibit both *cis-* and *trans-*bridging modes, which is different from previously reported polynuclear cobalt complexes and lead-transition metal $\beta$-diketonate coordination polymers. With high volatility and solubility, this heterometallic compound with unprecedented transition metal:lead ratio can serve as a precursor for the low-temperature preparation of prospective functional thin film materials such as lead–cobalt composite coating, lone-pair multiferroics, and superconductors. Importantly, this unique molecule with two distinctively different cobalt positions should be investigated as a model structure for the design of more complex hetero*tri*metallic compounds with partial substitution of cobalt with other divalent metals.

**Supplementary Materials:** The following supporting information can be downloaded at: https://www.mdpi.com/article/10.3390/cryst13071089/s1, Figure S1: DART-mass spectrum of [Pb$_2$Co$_5$(acac)$_{14}$] in a positive mode; Table S1: Assignment of ions detected in a positive-ion DART-mass spectrum of [Pb$_2$Co$_5$(acac)$_{14}$]; Figure S2: IR spectrum of [Pb$_2$Co$_5$(acac)$_{14}$]; Figure S3: Crystal structure of [Pb$_2$Co$_5$(acac)$_{14}$]; Table S2: Bond distances (Å) and angles (º) in the structure of [Pb$_2$Co$_5$(acac)$_{14}$]; Figure S4: Crystal structure of PbCo(acac)$_4$; Table S3: Bond distances (Å) and angles (º) in the structure of [PbCo(acac)$_4$].

**Author Contributions:** Y.Z. performed synthesis, obtained suitable single crystals for X-ray diffraction analysis, carried out characterization, and prepared the draft of the manuscript; Z.W. performed synchrotron single-crystal measurement and refinement, and provided structural description; E.V.D. conceived and supervised the project and finalized the manuscript. All authors have read and agreed to the published version of the manuscript.

**Funding:** This work was supported by the National Science Foundation (CHE-1955585).

**Data Availability Statement:** Data in this study are available upon request.

**Acknowledgments:** NSF's ChemMatCARS, Sector 15 at the Advanced Photon Source (APS), Argonne National Laboratory (ANL) is supported by the Divisions of Chemistry (CHE) and Materials Research (DMR), National Science Foundation, under grant number NSF/CHE-1834750. This research used resources of the Advanced Photon Source, a US Department of Energy (DOE) Office of Science user facility operated for the DOE Office of Science by Argonne National Laboratory under Contract No. DE-AC02-06CH11357.

**Conflicts of Interest:** The authors declare no conflict of interest.

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
