# Peer review of "Synthesis, Structure, and Characterizations of a Heterobimetallic Heptanuclear Complex [Pb2Co5(acac)14]"

_crystals, doi:10.3390/cryst13071089_

Round 1

Reviewer 1 Report

Dikarev et al. reported on an unusual heterobimetallic heptanuclear complex with Pb(II), Co(II) and acetylacetonate, which was synthesized by the gas phase/solid-state technique. Its composition seems quite unique among such types of complexes with β-diketonate ligands. The crystal and molecular structure of this new complex was nicely and comprehensively described, also in terms of chirality, wich is highly appreciable. It is also interesting that, due to the high-volatility of such heterobimetallic complex, it can be used as a precursor for the MOCVD technique at low temperature to obtain new thin film nanostructured functional materials. Overall, the work can be accepted for publication in Crystals with no further revision.

Author Response

Dear Reviewer, 

Thanks for your comments! 

Sincere, 

Yuxuan Zhang 

Prof. Dikarev's research lab

Reviewer 2 Report

In the manuscript submitted by Zhang et. al., discusses the synthesis and structure of one heterometallic heptanuclear molecular complex [Pb2Co5(acac)14]. The complex is new and interesting: It has a unique stoichiometry and a unique structure. Moreover, the complex is volatile and may be of interest for obtaining thin films/coatings under MOCVD conditions.

However, the manuscript contains only preliminary information and cannot be published as a full paper (article).

1. The manuscript is very short and presents the synthesis and structure of only one compound. The structure of [Pb2Co5(acac)14] is discussed in detail, however, there is no explanation for the choice of conditions for the synthesis of the target compound. In particular, there is no discussion of IR spectra. The synthetic part does not provide elemental analysis data (C and H), which prove the purity of the obtained compound, the yield of the target product under conditions of solid-state and solution synthesis is not indicated. Mass spectra are given in SI, but they are not commented on or discussed in any way.

2. Fig. 3 shows the structure of an already known compound (see reference [6]). This figure is best placed in SI.

3. Figure 4 shows the crystal structure of [PbCo(acac)4]. The synthesis of this compound and the results of SCXRD analysis are given in SI. It is not clear whether it is a new or already known compound? Differences in the synthesis conditions of [PbCo(acac)4] and [Pb2Co5(acac)14] are not discussed.

4. An interesting property of [Pb2Co5(acac)14] is its volatility. However, this important property has not been studied in depth.

Author Response

Dear Reviewer, 

Sincere, 

Yuxuan Zhang

Reviewer 3 Report

The manuscript by E.V. Dikarev and co-workers reports the synthesis of tnew volatile bimetallic heptanuclear complex [Pb2Co5(acac)14] by two alternate methods, i.e., solid state sublimation and reaction in the solution. It was characterized by single crystal X-ray diffraction, PXRD, FTIR, DART-MS, and ICP-OES analysis.

This paper may be interesting for the readers of Crystals. However, some revision is needed before its publication. The referee’s comments are given below.

1. The first paragraph in the 3.1 section should be transferred in the Experimental section.

2. How about the novelty of both synthesis procedures. If they are already known, relevant citations should be done in the paper.  

3. What is a product yield of the [Pb2Co5(acac)14] complex? How it was isolated in both procedures and separated from unreacted Pb- and Co-precursors?

4. In Abstract in Please specify the unique characteristics except Co/Pb ratio of the [Pb2Co5(acac)14] complex, which are different from other PbxMy(β-dik)2x+2y complexes.

5. This paper is too short. The structural analysis only for the complex by SCXRD and PXRD is considered in the manuscript. Please add an appropriate discussion of the obtained results of physico-chemical characterization by other methods.

6. Please supplement the Introduction and especially Conclusion with a more outlook about possible application of the bimetallic complex. Otherwise, this paper will be interesting for narrow readership specialized in the synthesis of the metal complexes. As to potential application as lead-based functional materials, e.g., anodes of [Pb2Co5(acac)14] complex, what are their advantages of? Obviously, lead-based materials are dangerous for environment.

7. “Heterobimetallic complex” – Heterometallic or bimetallic complex is more accurate name.

English language in this paper is quite good.

Author Response

Dear Reviewer, 

Sincere, 

Yuxuan Zhang

Reviewer 4 Report

The paper reports a new molecular Pb-Co complex with acac ligands. The compound has a unique Co:Pb = 5:2 ratio, which was clearly indicated by ICP-OES analysis and single-crystal X-ray diffraction analysis. The authors obtained this compound not only from regular solution synthesis but also from solid-gas phase synthesis. Although several Co-Pd-acac coordination compounds have been reported, this compound's unique composition and elimination nature is potentially interesting for readers of Crystals. So, I recommend this paper for publication after considering the following very minor issues.

- Figure 1 showed experimental and simulated PXRD patterns from the solution experiments. Please add PXRD pattern of the sample obtained from the solid-phase reaction in supporting information to indicate the phase purity of the solid-reaction product.

- Regarding “temperature gradient of 130”, indicate the unit of temperature gradient.

Author Response

Dear Reviewer, 

Sincere,

Yuxuan Zhang

Round 2

Reviewer 2 Report

The reviewer is not satisfied with the responses to comment 1. Other changes made are helpful.

Authots: What kind of information can we get from the C, H elemental analysis? That we have 14 acac ligands in the structure, so that all acac positions are fully occupied? First of all, we can see that from crystallography.

Reviewer : It is obvious to this reviewer that all new compounds must be characterized by elemental analysis. These data, in combination with other methods, prove the phase purity of the new compound. The samples obtained by the authors may contain impurities, in particular, X-ray amorphous ones. Figure 2 shows X-ray powder diffraction patterns of [Pb2Co5(acac)14] bulk product obtained from solution reaction and the Le Bail fit. To prove phase purity, the reviewer would like to see the calculated diffraction pattern and the original experimental diffraction patterns in a much larger range up to 2 theta 45 deg.

Authots: Second, if some acac ligands are missing, you suggest that we have Pb(I) or Co(I) ions in the structure?

Reviewer :No, I do not think so. The reviewer is familiar with basic inorganic chemistry and the chemistry of coordination compounds. This reviewer does not consider the possibility of the formation of Pb(I) or Co(I) ions in the structure.

Authots: Third, from 300+ structures of heterometallic diketonates, it has never been detected any partial occupation of the ligand.

Reviewer :The reviewer is confident that most of these 300+ complexes have been characterized by elemental analysis data.

Authots: Finally, some careless work on the analyzed moisture-sensitive sample can lead to partial hydrolysis/solvolysis and inaccurate data. Instead, we have run ICP-OES analysis to confirm the Pb:Co ratio of 2:5, which only matters in this case. Please, note that we are talking about the ratio, not the element content, and that is not depended on the handling of the sample.

Reviewer : The ratio of the elements does not contradict the data of X-ray diffraction analysis, but it does not prove the purity of the obtained compound either.

Author Response

Dear Reviewer, 
